# Bimetallic M–Cu (M = Ag, Au, Ni) Nanoparticles Supported on γAl_2_O_3_-CeO_2_ Synthesized by a Redox Method Applied in Wet Oxidation of Phenol in Aqueous Solution and Petroleum Refinery Wastewater

**DOI:** 10.3390/nano11102570

**Published:** 2021-09-30

**Authors:** Zenaida Guerra-Que, Jorge Cortez-Elizalde, Hermicenda Pérez-Vidal, Juan C. Arévalo-Pérez, Adib A. Silahua-Pavón, Gerardo E. Córdova-Pérez, Ignacio Cuauhtémoc-López, Héctor Martínez-García, Anabel González-Díaz, José Gilberto Torres-Torres

**Affiliations:** 1Centro de Investigación de Ciencia y Tecnología Aplicada de Tabasco (CICTAT), DACB, Laboratorio de Nanomateriales Catalíticos Aplicados al Desarrollo de Fuentes de Energía y Remediación Ambiental, Universidad Juárez Autónoma de Tabasco, Km. 1 carretera Cunduacán-Jalpa de Méndez, C.P., Cunduacán 86690, Tabasco, Mexico; LINK-190@hotmail.com (J.C.-E.); vidal_conde@hotmail.com (H.P.-V.); carlos.arevalo@ujat.mx (J.C.A.-P.); adibab45@gmail.com (A.A.S.-P.); enrique_cordova90@hotmail.com (G.E.C.-P.); nachoftir@gmail.com (I.C.-L.); hectorujat@gmail.com (H.M.-G.); 2Laboratorio de Investigación 1 Área de Nanotecnología, Tecnológico Nacional de México Campus Villahermosa, Km. 3.5 Carretera Villahermosa–Frontera, Cd. Industrial, C.P., Villahermosa 86010, Tabasco, Mexico; 3Laboratorio de Análisis y Caracterización, Universidad Juárez Autónoma de Tabasco, DAIA, Km. 1 Carretera Cunduacán-Jalpa de Méndez, C.P., Cunduacán 86690, Tabasco, Mexico; anaiq86@hotmail.com

**Keywords:** alloy nanoparticles, bimetallic catalysts, metal leaching, mineralization process, metal–metal interaction, reducibility, active oxygen species

## Abstract

Three bimetallic catalysts of the type M–Cu with M = Ag, Au and Ni supports were successfully prepared by a two-step synthesized method using Cu/Al_2_O_3_-CeO_2_ as the base monometallic catalyst. The nanocatalysts were characterized using X-ray diffraction (XRD), temperature-programmed reduction of H_2_ (H_2_-TPR), N_2_ adsorption-desorption, scanning electron microscopy (SEM) with energy dispersive X-ray spectroscopy (EDS), transmission electron microscopy (TEM) and ultraviolet–visible spectroscopy with diffuse reflectance (DR-UV-Vis) techniques. This synthesized methodology allowed a close interaction between two metals on the support surface; therefore, it could have synthesized an efficient transition–noble mixture bimetallic nanostructure. Alloy formation through bimetallic nanoparticles (BNPs) of AgCuAlCe and AuCuAlCe was demonstrated by DR–UV–Vis, EDS, TEM and H_2_-TPR. Furthermore, in the case of AgCuAlCe and AuCuAlCe, improvements were observed in their reducibility, in contrast to NiCuAlCe. The addition of a noble metal over the monometallic copper-based catalyst drastically improved the phenol mineralization. The higher activity and selectivity to CO_2_ of the bimetallic gold–copper- and silver–copper-supported catalysts can be attributed to the alloy compound formation and the synergetic effect of the M–Cu interaction. Petroleum Refinery Wastewater (PRW) had a complex composition that affected the applied single CWAO treatment, rendering it inefficient.

## 1. Introduction

Degradation of refractory organic compounds (ROCs) is a means of efficiently treating industrial wastewater that is released by the petrochemical, leather, coloring, textile or other related industries [1,2,3]. Despite the effort applied during the biological treatment of the industrial effluent, the results are not as desired. It is well known that ROCs are resistant to conventional treatment due to their low biodegradability. The low biodegradability of ROCs is caused by their high toxicity and low solubility in water [4,5,6]. In this work, phenol has been selected as a target compound, because it is one of the ROCs registered on the US EPA list of priority hazardous substances [7]. Phenol is able to be generated by petroleum refinery wastewater (PRW) [8]. Oil and grease represent a class of pollutants with very low affinity to water, very low biodegradability, high concentration and variability in ROCs composition, so they are released into the environment via PRW, and this may have an impact on the biosphere [9,10,11]. Consequently, it is challenging to ensure the high conversion of PRW into carbon dioxide and water or into products that can be eliminated by biological treatment after reducing the concentration of these pollutants to be treated further. An efficient strategy is urgently needed to achieve total mineralization of any refractory organic molecules, which is hindered by the increasingly stringent environmental regulations.

In order to achieve efficient degradation of ROCs, among many other processes, catalytic wet air oxidation (CWAO) is a technology that has been used to treat wastewater containing organic compounds that are highly toxic or too concentrated to be treated only with biological treatment [12]. The process involves oxidizing the ROCs under oxygen pressure at an elevated temperature and in the presence of a catalyst, which can take the form of metal transition oxides or noble metals, either heterogeneous or homogeneous catalysts. The use of a catalyst significantly improves the conditions of the process, so CWAO can be operated at temperatures and pressures below 200 °C and 30 bar, respectively [2,13,14]. The process can be directed in two different ways. The first involves full mineralization of the organic pollutants into CO_2_, N_2_, H_2_O and mineral salts and the second involves obtaining an increase in the effluent’s biodegradability. The last method involves the conversion of the toxic organic compounds towards the formation of more biodegradable products or to significantly reduce the concentrations of these pollutants [1,6].

Many heterogeneous catalytic systems have been developed for CWAO treatment of ROCs; among them, transition and noble metal catalysts containing Pt, Ru, Ni and Cu nanoparticles over different supports have been reported [6,15,16,17,18,19]. The composition and structure of heterogeneous catalysts play an important role in maximizing the efficiency of the degradation of ROCs [20,21].

However, the high price of noble metals restricts their use in CWAO of ROCs, so it is necessary to employ a cheaper alternative, such as transition metals or mixed nanoparticles of noble and transition metals with a low content of noble metal. Furthermore, much research has been conducted about mineralization using the addition of metal-based or transition catalysts such as copper and nickel due to their improved stability in complex systems and their low cost compared to noble metals [5,12,16,22].

ROC oxidation using Cu/CeO_2_ [15], Cu/CeO_2_ [5], Cu/Al_2_O_3_ [23] and Cu-CeO_2/_γAl_2_O_3_ [23] has been reported at 90–180 °C, and the total organic carbon (TOC) removal efficiency of the copper-supported catalyst was 78, 70, 74.5 and 84.6%, respectively [5,15,23]. Despite the good catalytic performance of copper, under oxidation conditions, it can be leached into the reaction matrix because it suffers from poor stability in an acidic medium. The degree of copper leaching has been found to be favorably high at pH<4; the formation of acidic intermediates such as short-chain carboxylic acids has led to a decline in the pH value during the CWAO of ROCs [24,25].

Precisely, the addition of a second metal to form a bimetallic system has significantly enhanced the stability and the activity of the designed catalysts due to a synergistic real interaction [26,27]. As we know, the nanostructured nanomaterials must be designed with high stability under conditions of leaching and agglomeration or sintering during the reaction in order to achieve high catalytic performance [5,28].

It is well known that the preparation methods of bimetallic catalysts appropriately carried out can strongly improve the stability of the catalyst based on the synergy of the two active phases [29,30]. It is a challenge to improve the stability of the first metal by alloying it with the addition of a second metal. However, our group decided to evaluate a general two-step method called the redox method in order to promote a real metal–metal interaction, involving copper–silver, copper–gold and copper–nickel, and to establish which interaction is more beneficial for the activity of CWAO of phenol and PRW.

Indeed, our previous studies have shown that the redox method is an efficient method that promotes a close interaction between two metals on the support surface [31]. It is based on the reduction of the ions of the second metal by a reducing agent adsorbed at the surface of the pre-reduced first metal—for instance, hydrogen. The redox method can ensure an extended area of contact between the metallic phases; this phenomenon is favored by the surface interaction of the primary supported component with a precursor of the second metal [32,33,34].

When looking for a close interaction between M and Cu, therefore improving copper’s stability, which additionally appropriately increases the catalytic performance of CWAO of phenol and PRW, we tested different metals over copper, such as noble-based or transition metals.

Indeed, most recently, catalysis research has focused on the addition of transition metals because they are considered in the research field as low-cost catalysts, when they are a transition–noble mixture bimetallic nanostructure. The bimetallic catalysts synthesized and tested in CWAO of phenol and PWR under mild conditions by our research group were Ag-Cu/Al_2_O_3_–CeO_2_ (AgCuAlCe), Au-Cu/Al_2_O_3_–CeO_2_ (AuCuAlCe) and Ni-Cu/Al_2_O_3_–CeO_2_ (NiCuAlCe).

The present research focuses on comparing Ag-Cu, Au-Cu and Ni-Cu versus Cu supported on Al_2_O_3_-CeO_2_. COD and TOC changes were tested to clarify the exact effect of bimetallic NPs on the conversion of phenol and petroleum refinery wastewater by CWAO under mild conditions (120 °C and 10 bar O_2_). Furthermore, another important contribution was to improve the reaction conditions at temperatures and pressures as mild as possible, from an economical point of view. The synthesized catalysts were also characterized by different techniques, such as X-ray diffraction (XRD), temperature-programmed reduction of H_2_ (TPR), N_2_ adsorption–desorption, scanning electron microscopy (SEM) with energy-dispersive X-ray spectroscopy (EDS), transmission electron microscopy (TEM) and diffuse reflectance ultraviolet–visible spectroscopy (DR–UV–Vis) to study the influence of the formulation on their textural properties and activity.

## 2. Materials and Methods

### 2.1. Support Preparation

The alumina (Al_2_O_3_) and Al_2_O_3_-CeO_2_ supports were prepared by the sol–gel method. The Al_2_O_3_ support was prepared by using aluminum tri-sec-butoxide precursor salt solution (from Aldrich) dissolved in water at pH 3 using acetic acid (CH_3_COOH). A mixture of n-butanol–water was stirred and kept at room temperature. Aluminum tri-sec-butoxide was added drop by drop for 3 h to the solution described above until a gel was formed. The mixture was constantly stirred for 24 h at room temperature. Afterwards, the water and alcohol remaining were eliminated using a rotavapor unit. Then, the powder obtained was left in an oven to dry at 120 °C for 12 h. The samples were calcined at 500 °C for 12 h with heating ramp of 2 °C/min.

The Al_2_O_3_-CeO_2_ support was obtained by using cerium nitrate precursor salt (from Aldrich). Cerium aqueous solution were obtained by the stoichiometric addition of precursor to obtain 5 wt% CeO_2_. For Al_2_O_3_-CeO_2_, the same methodology used to obtain the Al_2_O_3_ was followed, and the precursor salt was added to the n-butanol–water mixture before adding it to the solution of aluminum tri-sec-butoxide–water.

### 2.2. Monometallic and Bimetallic Catalyst Preparation

The copper was loaded on the respective support (Al_2_O_3_-CeO_2_) by wet impregnation (10 g) with aqueous solution of copper nitrate containing the required amount to obtain a nominal concentration of 5% of Cu with urea under stirring for 24 h at room temperature. It was taken as the base 10 mL of total solution/g support. When urea was used in the catalyst’s preparation, the Cu:urea molar ratio was 1:1 and the pH of the impregnation solutions was adjusted to 10 with aqueous sodium hydroxide. After impregnation, catalysts were dried at 120 °C for 12 h and then calcined under air flow (60 mL/min) at 500 °C for 5 h, with a heat rate of 2 °C min^−1^. Finally, the monometallic catalysts were reduced under H_2_ (60 mL min^−1^) at 400 °C for 4 h, with a heat rate of 2 °C min^−1^.

The bimetallic catalysts were prepared by the recharge method, reducing AuCl_4_^−^, Ag^1+^, Ni^2+^ (from HAuCl_4_, AgNO_3_ and Ni(NO_3_)_2_.6H_2_O) with pre-adsorbed hydrogen on the copper-supported surface. An amount of 2 g copper monometallic catalyst supported on mixed oxides of alumina–ceria was introduced into a reactor under nitrogen flow and was activated at 400 °C for 1 h under a hydrogen atmosphere. Next, the solution of the gold, silver or nickel precursor, previously degassed under a stream of nitrogen, was introduced onto the catalyst, taking an amount sufficient to synthesize a 1:1 molar ratio. After a reaction time of 1 h under hydrogen bubbling at room temperature, the bimetallic catalyst was dried with hydrogen at room temperature, then at 100 °C (heating rate 2 °C min^−1^) overnight. Finally, the three bimetallic catalysts synthesized were reduced under a stream of hydrogen at 400 °C for 1 h, with a heating rate of 2 °C min^−1^.

### 2.3. Reaction Conditions

The activity testing of the catalysts synthesized in this study was carried out in a 300 mL Parr batch reactor, with the following conditions: 120 °C, 10 bar and 1000 ppm of phenol. The standard procedure for a CWAO experiment was followed, 250 mL of pollutant solution was poured, and 0.25 g of catalyst was applied in the 300 mL reactor. When the selected temperature was reached, stirring started at a maximum speed of 1000 rpm. This time was taken as the zero-reaction time and the reaction duration was 180 min. These conditions were the same for all the synthesized materials. The liquid samples were periodically removed from the reactor, then filtered to remove any catalyst particles and, finally, analyzed by gas chromatography, total organic carbon and chemical oxygen demand.

Conversion values of chemical oxygen demand (COD) [35], total organic carbon (TOC) and phenol were determined using the following equations at different times of 30 min intervals up to 180 min of reaction:(1)$$X_{COD}=\frac{COD_{0}-COD_{180}}{COD_{0}}\times 100\%$$
(2)$$X_{TOC}=\frac{{\mathrm{TOC}}_{0}-{\mathrm{TOC}}_{180}}{{\mathrm{TOC}}_{0}}\times 100\%$$
(3)$$X_{Phenol}=\frac{C_{0}-C_{180}}{C_{0}}\times 100\%$$
where TOC_0_ is TOC at t = 0 (ppm), *COD*_0_ is *COD* at t = 0 (ppm), C_0_ is phenol concentration at t = 0 (ppm), C_180_ is phenol concentration at t = 3 h of reaction (ppm), TOC_180_ is TOC at t = 3 h of reaction (ppm), *COD*_180_ is *COD* at t = 3 h of reaction (ppm). Furthermore, the selectivity was calculated according to the following equation.
(4)$$S_{CO_{2}}=\frac{X_{TOC}}{X\;Phenol}\times 100\%$$

Initial rate (*r_i_*) [36] was calculated from the conversion of phenol in accordance with time, using the following equation:(5)$$r_{i}=(\frac{\Delta_{Phenol}\left(\%\right)}{\Delta tm_{cat}})\;({\left[pollutant\right]}_{i})$$
where $\frac{\Delta_{Phenol\;}\left(\%\right)}{\Delta t}$ is the initial slope of the conversion curve; [*pollutant*]*_i_* = initial phenol concentration and *m_cat_* = catalyst mass (g_cat_ L^−1^).

#### 2.3.1. Gas Phase Chromatography

The GC analysis was carried out in a PerkinElmer gas chromatograph with a flame ionization detector. The temperature of the injection port and detector was maintained at 200 and 275 °C, respectively. The injection volume was 1 μL. The column was a VF-1ms with dimensions of 30 m (Length), 0.25 mm (ID) and 0.25 μm (Film Thickness). The oven temperature was maintained at 180 °C (isothermal treatment). The carrier gas used was helium of 99.999% purity.

#### 2.3.2. Chemical Oxygen Demand

The COD of the phenol solutions treated by CWAO was determined by potassium dichromate standard, with a colorimetric method (5220D). The HACH vials contained the digestion solution (K_2_Cr_2_O_7_) and the sulfuric acid reagent (H_2_SO_4_) to measure COD at the range of 0–1500 mg L^−1^. At the beginning, 2 mL of sample or blank was added, with at least five standards from potassium hydrogen phthalate solution with COD equivalents to cover the concentration range. Afterwards, the prepared vials were placed in an oven preheated to 150 °C for 2 h to digest the organic matter. Finally, we measured the absorption of each sample, blank and standard, at a selected wavelength (600 nm). The effectiveness of the catalysts was determined in terms of percentage COD removal.

#### 2.3.3. Total Organic Carbon (TOC)

Total organic carbon of the initial and final solution of the pollutant (phenol and PWR), at the beginning and at the end of the catalytic test, was measured using a Shimadzu analyzer TOC-L CSN. Total carbon (TC) and inorganic carbon (IC) were determined separately; finally, TOC was calculated by subtracting IC from TC. The determination of CO_2_ was carried out by non-dispersive infrared detection (NDIR). The effectiveness of the catalysts was determined in terms of percentage TOC removal.

### 2.4. Characterization Techniques

#### 2.4.1. BET Specific Surface Area ($S_{BET}$)

Nitrogen physisorption was used to establish the isotherms of adsorption, the distribution of pore size and the specific surface area. The surface areas of the samples were determined from the nitrogen adsorption isotherms at −196 °C in a MicromeriticsTristar 3020 II. Before analysis, the samples were degassed at 400 °C for 4 h. The adsorption data were analyzed using the ASAP 2020 software based on the Brunauer–Emmett–Teller isotherm (BET).

#### 2.4.2. X-ray Diffraction (XRD)

X-ray diffraction (XRD) analysis was used to determine the phase composition and to estimate the crystallite size of the powders. XRD was carried out using a Bruker D2 PHASER diffractometer with radiation source Co Kα (λ = 0.179 nm) with an analysis time of 650 s. The average crystal size in the supports and catalysts was estimated using the Scherrer equation [31]:(6)$$D=\frac{0.90\lambda }{\beta cos\theta }$$
where *D* is the crystal size (nm), λ is the wavelength (nm), *β* is the corrected full width at half maximum (radian) and *θ* is the selected diffraction angle (radian).

An additional equation was also used to calculate the average oxide crystal size ($d_{S_{BET}}$) [1]. The $d_{S_{BET}}$ was calculated from $S_{BET}$, assuming that the particles were semicrystalline spherical:(7)$$d_{S_{BET}}=\frac{6\;V}{S_{BET}}$$
where *V* is the specific volume of the oxide or metal (m^3^ g^−1^) and $S_{BET}$ is the specific surface area of the oxide or metal (m^2^ g^−1^).

#### 2.4.3. Ultraviolet–Visible Spectroscopy with Diffuse Reflectance (DR–UV–Vis)

The UV–Vis spectra with diffuse reflectance at the range of 200–900 nm with a diffuse reflectance accessory (integration coupled to sphere) were obtained with a Varian Cary 3000 spectrometer that functioned at room temperature. The BaSO_4_ compound was used as a reference with 100% reflectivity, to establish the baseline.

#### 2.4.4. Scanning Electron Microscopy (SEM)

Samples of the bimetallic catalysts supported on Al_2_O_3_-CeO_2_ were analyzed by scanning electron microscopy (SEM). Samples were mounted on double-sided carbon conductive tape in an aluminum sample holder for morphological analysis. Later, they were observed in a scanning electron microscope, JEOL JSM-6010LA. The characteristics of the analysis were 20 kV acceleration voltage under high vacuum conditions at 5000X and 35000X. An energy-dispersive X-ray spectrometer detector (EDS) coupled to SEM was used to perform the semiquantitative analysis and distribution of elements on the surface of the samples. The images were processed in the InTouchScopeTM Software Version 1.03A (JEOL TECHNICS LTD). More than 300 particles were selected to estimate the average diameter value of bimetallic nanoparticles (NPS).

The particle average diameter (*dm*) was calculated using the formula:(8)$$dm=\sum_{i}(x_{i}d_{i})/\sum_{i}x_{i}$$
where *x_i_* is the number of particles with diameter *d_i_*.

#### 2.4.5. Transmission Electron Microscopy (MET)

Transmission electron microscopy (TEM) was performed in a JEOL JEM2100 STEM. The samples were ground, suspended in ethanol at room temperature and dispersed with agitation in an ultrasonic bath for 15 min; then, an aliquot of the solution was passed through a carbon copper grid. The particle size distribution of the catalysts was obtained by measuring more than 50 nanoparticles in each sample.

#### 2.4.6. Temperature-Programmed Reduction of H_2_ (H_2_-TPR)

The H_2_-TPR experiments were performed on Belcat equipment with a thermal conductivity detector, using 0.05 g of catalyst. Samples were previously treated with the following protocol: Ar flow for 55 min at 130 °C, Ar flow for 16 min at 35 °C. Subsequently, for the H_2_-TPR analysis, the temperature was raised from room temperature to 500 °C at a heating rate of 10 °C/min with a flow of 5% H_2_/Ar for one hour. The same procedure was applied to the samples with nickel content; only one change was made—the temperature of H_2_-TPR for the analysis was raised to 900 °C.

## 3. Results

### 3.1. Material Characterization

#### 3.1.1. BET Specific Surface Area (*S_BET_*)

N_2_ physisorption experiments were performed to determine the $S_{BET}$, the total volume and pore size. Indeed, the specific surface area and porosity play a crucial role in promoting the diffusion and transport of molecules to the active sites of the heterogeneous catalyst in the oxidation reaction. Figure 1 shows the graphs of the N_2_ adsorption–desorption isotherms of the monometallic and bimetallic copper-supported catalysts.

The graphs of the N_2_ adsorption–desorption isotherms of the catalysts showed the hysteresis phenomenon at the relative pressure range 0.55–0.85 (P/PO). The hysteresis loops situated in the indicated range were a consequence of a mesoporous structure existing in the synthesized materials [37].

All bimetallic catalysts including monometallic catalysts had regular porous networks of isotherms type IV with narrow H1-type hysteresis loops, which were associated with capillary condensation, typically presented in mesoporous materials, as defined by IUPAC [38,39].

On the other hand, the hysteresis cycle of the AuCuAlCe bimetallic catalyst displayed a slight change towards a lower relative pressure, showing that its pore diameter was smaller than that of the rest of the counterpart bimetallic-based copper catalysts.

Table 1 lists the textural properties of the support, monometallic and bimetallic catalysts. The $S_{BET}$ of the bimetallic catalysts had slightly lower values than the monometallic catalyst. However, another important observation pertains to the bimetallic catalyst: it was within the set of supported catalysts, and NiCuAlCe had $S_{BET}$ slightly higher compared to the other counterparts. The addition of the second metal—in other words, Ag, Au and Ni—in the copper monometallic-supported catalyst caused a decrease in the $S_{BET}$ in the CuAlCe, which indicated that the second metals were possibly deposited on the internal surface of the monometallic catalyst, or that they caused the collapse of the porous structures of AlCe. The pore sizes of these bimetallic catalysts were within the range of 7–9 nm; these findings support the notion that they are mesoporous materials [40].

#### 3.1.2. X-ray Diffraction (XRD)

The X-ray diffraction patterns of the monometallic and bimetallic-supported catalysts were used to identify the catalysts’ phases, are shown in Figure 2. It was observed that after second metal addition over the monometallic copper-supported catalyst by the redox method, the three synthesized bimetallic catalysts preserved the support structure, namely the γ structures of the alumina (PDF-01-075-0921, PDF-00-004-0858)—that is, the face-centered cubic (FCC) structure of O^2−^ ions with vacancies of Al^3+^—and also preserved the FCC ceria fluorite structure (PDF-03-065-7999, PDF 01-071-4199) of Ce^4+^ cations with possible vacancies of O^2−^. However, this can be seen in our additional findings in AgCuAlCe, AuCuAlCe and NiCuAlCe. The diffraction pattern of the AgCuAlCe bimetallic catalyst showed a silver metallic phase (PDF-00-0001-1164, PDF-00-003-0921), copper metallic phase (PDF-00-003-1005, PDF-00-001-1241) and oxidized copper species, Cu_2_O (PDF-01-071-3645). The diffraction pattern of the AuCuAlCe bimetallic-supported catalyst showed characteristic and intense peaks of metallic gold (PDF-00-004-0787, PDF-01-071-3755, PDF-01-071-4616, PDF-03-065-2870), besides a less intense peak of Cu^0^.

The Ni^0^ peaks were identified with the files, PDF-00-001-1266, PDF-00-001-1258, PDF-00-004-0850. These files indicate the FCC structure with space group Fm-3 m. It should be noted that the diffraction peaks due to the cubic Ni phase were overlapped with the rather broad and short diffraction peaks of the Cu phase in the XRD patterns of the NiCuAlCe bimetallic catalyst. It is difficult to discern the detection of XRD signals due to the cubic Ni phase.

The average particle size obtained using Scherrer’s equation (Table 2) showed that, when the width of the peak is smaller, there is a larger particle size and vice versa. As a result, larger silver particles (14 nm) and smaller gold particles (9 nm) were found.

These results suggest that copper in the three bimetallic catalysts was finely dispersed on the alumina–ceria surface.

#### 3.1.3. Ultraviolet–Visible Spectroscopy with Diffuse Reflectance (RD–UV–Vis)

Diffuse reflectance UV–Vis spectroscopy is a useful technique to characterize bimetallic nanoparticles (BNPs). The electronic spectra of bimetallic-supported catalysts are shown in Figure 3a,b, respectively. Figure 3c depicts the DR–UV–Vis spectra of monometallic NPs of supported copper on Al_2_O_3_-CeO_2_.

The spectrum of monometallic CuAlCe revealed a broad band that was centered near 650 nm and was attributed to Surface Plasmon Resonance (SPR) of copper NPs [41]. Moreover, the SPR of gold NPs reported in the literature is located near 530 nm [42,43,44]. Figure 3a shows a short and weak peak centered near 570 nm for AuCuAlCe. This can be associated with the SPR of AuCu-supported BNPs. However, the particle size, shape and support interaction have been found to influence the shape and position of the SPR band of metallic and bimetallic NPs [43,45].

In addition, the SPR of monometallic silver NPs was reported to be 480 nm [42,46]. The short and weak peak at 600 nm of the UV–Vis spectra shown in Figure 3b can be assigned to the SPR of AgCu-supported BNPs.

The spectra of the bimetallic copper-based samples synthesized in this study were quite different from those of the pure metal copper-supported sample. The presence of one single absorption band of SPR of AgCu- and AuCu-supported BNPs, which was located at the intermediate space between that of the monometallic one, verified that the particles were not a mixture of separated monometallic particles but constituted of truly alloy-formed BNPs. This evidence was associated with the formation of alloy BNPs. The results agree with previous reports [38,42,45,46,47].

#### 3.1.4. Scanning Electron Microscopy (SEM)

Three different bimetallic-supported systems were prepared with a redox method using a specific salt precursor for each metal. The particle sizes and morphologies of the fresh samples were determined by SEM. The SEM images of AgCu supported on AlCe are shown in Figure 4a,b. We observed smaller and smoother particles that were not uniformly dispersed on the support surface. The morphological structure showed irregular aggregates that were distinctly anchored and almost uniformly dispersed on the surface of AlCe and mostly exhibited a “sphere-like” morphology with an average diameter of 36 nm.

The presence of Ag and Cu NPs was detected (Figure 5). It showed an apparent composition of the AgCuAlCe bimetallic catalyst as determined by EDS analysis. Furthermore, the EDS results of AgCuAlCe indicated the greater availability of silver compared to copper on the catalyst surface.

The element mapping image (Figure 6) demonstrated that bimetallic particles were formed. The EDS elemental mapping analysis clearly revealed the existence of Ag, Cu, Al and Ce, indicating that Ag and Cu NPs were effectively deposited on the AlCe surface in a 350 µm^2^ area.

To verify that the second metal addition was successful, we performed a close EDS analysis over AgCuAlCe, results of which can be seen in Figure 7. The bright spots were confirmed as silver particles. There was a relationship between agglomeration and the intensity of the signal; in other words, silver agglomerated particles showed high intensity and highly dispersed silver particles showed the opposite in the elemental mapping of silver.

SEM micrographs of the fresh NiCuAlCe-supported nanomaterial are shown in Figure 8. The particle morphology of NiCuAlCe exhibited a “sphere-like” morphology with an average diameter of 74 nm. Furthermore, the scanning electron micrographs (Figure 8) showed that the particles obtained were few and were deposited on the surface of the support, and there were very few large and isolated particles formed separately.

The EDS spectrogram in Figure 9 shows the presence of Ni and Cu in this sample, without any significant variation in the theoretical content of both metals (Ni (4.59 wt%) and Cu (4.86 wt%)).

The elemental mapping of NiCuAlCe is shown in Figure 10. The result indicated that Ni and Cu atoms were mostly distributed uniformly over the entire set of NPs, but in the center of the sample, Cu atoms were agglomerated.

The average bimetallic particle size was estimated from the XDR peak using Scherrer´s equation and from SEM images through measurement of more than 300 particles (Table 2). The AuCuAlCe BNPs could not be seen on the surface of the support in the SEM images. This latter result is in good agreement with the XRD result.

#### 3.1.5. Transmission Electron Microscopy (TEM)

TEM images of fresh bimetallic-supported catalysts AuCuAlCe and AgCuAlCe can be seen in Figure 11, Figure 12, Figure 13 and Figure 14. For each sample, around 50 individual particles randomly selected in a unique zone of the catalyst were analyzed. The metal nanoparticles of gold and silver appeared darker in the images because they showed strong electron diffraction.

Figure 11 shows the TEM image and particle size distribution of AuCu bimetallic nanoparticles. The TEM image of the AuCuAlCe catalyst showed several agglomerated and many separated or isolated spherical BNPs distributed over the whole surface [2,47]. The average particle size of the Au was less than 6 nm, with a wide particle size distribution.

The medium magnification TEM image (Figure 12a) shows a larger nanoparticle with lattice fringes of 0.130 and 0.211 nm in region I (Figure 12b), which may fit with metallic Cu (220) and Cu (111), respectively, besides 0.123 and 0.249 nm, corresponding to metallic Au (311) and Au (111), respectively, values that are close to the results of lattice spacing from XRD.

In region II, the d spacing was found to be 0.207 nm, corresponding to the plane (220) of metallic gold [41,48]. Furthermore, the Al_2_O_3_ and CeO_2_ supports could be identified in region III, with interplanar spacing values of 0.261 nm for alumina and 0.312 nm for ceria, which were indexed to the (220) and (111) planes, respectively.

Furthermore, it was possible to indicate that Au and Cu over the Al_2_O_3_-CeO_2_ support existed in the form of bimetallic NPs with preferential segregation of Au in the surroundings of the particle. However, the formation of the Au-Cu alloy, monometallic Au and Cu, over the surface of the support could not be eliminated.

Figure 13a–c show TEM images of the AgCuAlCe bimetallic catalyst, confirming that the synthesized spherical isolated and agglomerated BNPs were well dispersed on the support. Analyses of the AuCuAlCe particle size distributions show a narrower distribution than AgCuAlCe, with most of the particle sizes ranging between 2 and 12 nm. Additionally, the larger particles were found to be of silver, whereas smaller particles were found to be of gold. Because the Au-Cu BNPs were highly dispersed in the support, it was possible to see less BNPs than Ag-Cu BNPs [49,50].

Figure 14b shows that the interplane distance in the lattice of Cu was 0.208 nm, which corresponds to the (111) plane of copper in the bimetallic catalyst. The lattice spacing values shown in Figure 14d are 0.235 and 0.205 nm, which correspond to (111) and (200) index planes of metallic Ag [51,52]. The individual and separate lattice fringes of silver and copper provide evidence of the core-shell structure in BNPs [52].

#### 3.1.6. Temperature-Programmed Reduction of H_2_ (H_2_-TPR)

H_2_-TPR profiles for the calcined and fresh monometallic and bimetallic catalysts are presented in Figure 15. The bimetallic catalysts were synthesized by the redox method and were analyzed to check the reduction temperature, to determine the number of reducible species and to investigate the reducibility of materials. The reduction pattern of the Cu monometallic-supported catalyst (Figure 15a) showed hydrogen consumption from around 60 to 260 °C, due to the reduction of copper oxide. In addition, the monometallic copper only showed three reduction peaks.

The AgCuAlCe bimetallic catalyst displayed major TPR at 60 °C, 138 °C, 22 5°C and 500 °C (Figure 15b). In contrast, AuCuAlCe (Figure 15c) exhibited a higher number of peaks than AgCuAlCe, due to the reduction of more different species of oxidized BNPs, at 75 °C, 238 °C, 310 °C, 363 °C and 495 °C. However, the H_2_-TPR profiles of NiCuAlCe (Figure 15d) revealed more reduction peaks at higher temperatures, centered at 100 °C, 240 °C, 300 °C, 400 °C, 650 °C and 850 °C.

The appearance of multiple peaks in the reduction profiles in the bimetallic catalysts can be associated with species of different characteristics, such as the degree of agglomeration and the strength of the metal–metal–support interaction. In other words, the differences in the reduction behavior of bimetallic catalysts AgCuAlCe (Figure 15a), NiCuAlCe (Figure 15c) and AuCuAlCe (Figure 15b) can be attributed to structural differences due to the heterogeneity of the BNP-supported surface that can be seen in the TPR profiles, with several peaks located at different positions and with different shapes. It is widely accepted that reduction peaks at lower temperatures are associated with the presence of bimetallic species that have a weak interaction with the support, such as “free” crystalline and the highly dispersed phase of small particles on the surface of the support; furthermore, peaks with higher temperatures correspond to bimetallic samples with a greater degree of agglomeration when in close contact or strong interaction with the support or isolated ion species of a specific metal [26,29,34]. Another important aspect of the reduction pattern is that TPR profiles with greater signal intensity suggest an increase in a particular species, in comparison to the others [27]. These latter results are in good agreement with the TEM result, as the TEM images showed a wide particle size distribution, especially in AgCuAlCe. Indeed, all synthesized bimetallic catalysts (Figure 15b–d) presented a greater number of reducible species than the copper monometallic catalyst (Figure 15a).

Redina et al. [27] and Mierczynski et al. [34] affirmed that the process of reducing a single oxidized bimetallic species occurs more easily when the peaks of reduction in a sample are located at lower temperature values in the specific peaks, compared to the reduction peaks of a monometallic material. The ease of the reduction process is likely to be one of the most important factors in determining catalyst performance [53].

The reduction profiles of Au, Ag and Ni monometallic-supported catalysts were reported in the literature [40,53,54]. In the case of Au/CeO_2_, it presented a reduction profile at 175 °C, 475 °C and 790 °C [53]. The TPR profile was observed at 292 °C, 425 °C, 606 °C, 697 °C and 852 °C for Ag/CeO_2_-Al_2_O_3_ [40]. Finally, the Ni/Al_2_O_3_-CeO_2_ reduction pattern was reported in previous work at 75 °C, 250 °C, 310 °C, 450 °C and 800 °C [54].

Furthermore, the two synthesized BNPs catalysts, Ag_2_O for AgCuAlCe and Au_2_O_3_ for AuCuAlCe, were reduced at lower temperatures than the respective monometallic catalyst, except with Ni species. The shift in the TPR maximum of Ag^1+^ and Au^3+^ to lower temperatures indicated that Cu catalyzes the reduction of the second metal species. This latter result highlights the mutual influence of the individual components on the reducibility of the BNP catalysts [55,56]. Addition evidence of possible alloy formation between M-Cu, where M can be Ag or Au, is given by the H_2_-TPR profiles, which confirmed the previous results seen in UV–Vis, SEM and TEM.

#### 3.1.7. Phenol Degradation through CWAO

CWAO experiments were carried out for the catalysts CuAlCe, AgCuAlCe, AuCuAlCe and NiCuAlCe under the following reaction conditions: initial phenol concentration of 1000 ppm, oxygen partial pressure of 10 bar and 120 °C. Adsorption of phenol onto the catalyst surface was determined for all the synthesized materials. The phenol conversion, COD and TOC were measured to compare the efficiency of the prepared catalysts in the CWAO.

Samples were withdrawn periodically during the reaction to determine the initial concentrations of phenol, COD and TOC, besides the final concentration after CWAO treatment. The effect of the second metal addition in the BNP catalyst for the CWAO of phenol is presented in Figure 16. Phenol was completely decomposed in 60 min using AgCuAlCe and 99% using AuCuAlCe, in contrast to 70% for NiCuAlCe.

Figure 17 depicts the COD and TOC removal efficiencies of phenol at 120 °C for 180 min in CWAO. The AuCuAlCe BNPs catalyst showed remarkable performance in phenol CWAO, as it reached a higher level of COD and TOC removal, with 89% and 93%, respectively, after 180 min of reaction. However, slightly lower COD and TOC removal was obtained for AgCuAlCe. In contrast, minimal mineralization (removal of TOC below 10%) was observed in the monometallic copper and NiCuAlCe bimetallic catalysts. The decrease in the COD and TOC indicated the progress of mineralization. In particular, the observed increase in COD removal indicated that the phenol molecule was oxidized to CO_2_ [4,15,55].

Therefore, within this group of bimetallic catalysts, the AuCuAlCe displayed the greatest efficiency of mineralization of Refractory Organic Matter, leading to the highest formation of CO_2_ and thus the highest selectivity in mild conditions (Table 3).

The mineralization can be explained by the conversion of the aromatic compounds into aliphatic compounds such as carboxylic acids (maleic acid, acetic acid, formic acid, oxalic acid) by the ring-opening reactions. In other words, the catalytic oxidation degradation pathway of phenol includes the hydroxylation (hydroquinone, catechol, o-benzoquinone, p-benzoquinone) and organic acids until CO_2_ and H_2_O are produced [4,23,57]. The mechanism including the formation of oxidized C_6_-aromatic of hydroquinone and catechol and the occurrence of both compounds indicates parallel reaction pathways [58]. The presence of catechol and hydroquinone may be attributed to hydroxyl radical attack at the ortho and para positions of the aromatic ring due to the resonance effect of phenol. In addition, from the above reactions, a solid residue may be also formed as a result of the combination of phenyl radical with hydroquinone and p-benzoquinone through a series of chain reactions [1].

Table 3 summarizes the catalytic oxidation of phenol over monometallic copper and BNPs catalysts. It was clear that the monometallic catalyst CuAlCe and BNP catalyst NiCuAlCe showed poor activity, while the BNP catalysts AuCuAlCe and AgCuAlCe exhibited high activity. The phenol oxidation reactions proceeded with distinct color and pH changes, displaying a soft brown color of the solution in the first 60 min, later changing from brown to colorless after 90 min.

It was possible to identify reaction products in these experiments through the GC, catechol, in lower concentrations for AgCuAlCe and AuCuAlCe than NiCuAlCe and CuAlCe. The above-mentioned finding corroborates the most notable AgCuAlCe and AuCuAlCe activity results compared to the other catalysts.

#### 3.1.8. Petroleum Refinery Wastewater Degradation through CWAO

The bimetallic catalysts were tested in the CWAO of petroleum wastewater in order to treat real industrial wastewater. The industrial stream used in this work was collected in a petroleum well assigned to Ecologica Transporte de Residuos Peligrosos. The generation process of industrial wastewater used in this study can be explained as released water from underground reservoirs to the surface during oil extraction. The main characteristics of the industrial wastewater sample are shown in Table 4.

The application of the experimental conditions to the industrial effluent was determined by the experimental design of the target molecule. Figure 18 shows that the results of COD abatement were 20%, 28% and 24% for NiCuAlCe, AgCuAlCe and AuCuAlCe, respectively. Before the treatment, the original sample showed 1909 mg L^−1^ of COD, and after CWAO treatment, it showed a value of 1527 mg L^−1^, 1374 mg L^−1^ and 1451 mg L^−1^, for NiCuAlCe, AgCuAlCe and AuCuAlCe, respectively. Furthermore, the measured TOC values were lower than COD for the three BNP catalysts (10%, 14% and 12% for NiCuAlCe, AgCuAlCe and AuCuAlCe, respectively).

The observed increasing trend of COD removal (NiCuAlCe < AuCuAlCe < AgCuAlCe) indicates that the mineralization process was started in the three bimetallic NP catalysts. However, the mineralization process generated intermediates only when sufficient formation of CO_2_ was achieved. Therefore, according to the TOC value, which is directly associated with the generation of CO_2_, AgCuAlCe showed better catalytic performance in the petroleum refinery under the selected conditions.

Petroleum wastewater can be generated in the processes of petrochemical industries and petroleum refineries (oil refining, oil extraction, oil storage, oil transportation). It is characterized by large volumes of oily wastewater with very low affinity to water, very low biodegradability, high toxicity and high hazard due to the high content of polycyclic aromatics [9,59,60]. The petroleum refinery wastewater generated by the extraction of crude oil had a mixture of a high concentration of aliphatic and aromatic petroleum hydrocarbons (benzene, toluene, ethylbenzene and xylene (BTEX)), polycyclic aromatic hydrocarbons (PAHs) and phenol, besides inorganic pollutants such as dissolved mineral salts (calcium and sodium chlorides, sodium carbonates, potassium chlorides, calcium or barium sulfates), sulfides, ammonia and heavy metals [9,10,61].

The results of COD and TOC value are inherent to the complexity of the industrial wastewater and the mild conditions selected in this study. For this reason, many organic wastewater treatment methods are associated with sequential techniques that involve several steps. These methods have been studied widely because of the potential to offset their individual disadvantages by means of adding adsorption technology, the Fenton/Fenton-like process, CWAO, photocatalytic oxidation, electrochemical processes and sonolysis technology [59,62].

## 4. Discussion

Bimetallic catalysts with noble metals have already been discussed; however, the high price of noble metals restricts their use in catalytic wet air oxidation (CWAO) of refractory organic compounds (ROCs). Therefore, here, we propose a combination of copper and another cheaper alternative such as a transition metal (nickel) or noble metal (silver or gold). The synthesis of a true nanoalloy catalytic-supported system with strong M–M interaction that simultaneously links one metal to another metal was achieved by a redox method. The principal characteristic of this method of preparation is the rearrangement of the surface, which leads to good dispersion at the surface of all the metals and a strong metal–metal interaction (Figure 19).

Copper monometallic supports have been leached into the reaction matrix due to poor stability; thus, in this research, we improve the stability of copper by alloying it with silver and gold. The redox method promoted in AgCuAlCe and AuCuAlCe such structural properties. The synergistic M–M effect was observed in the improvement of reducibility and catalytic performance of the wet air oxidation of a phenol aqueous solution. Indeed, reducibility is associated with structural defects in nanomaterials, such as oxygen vacancies or unsaturated sites, which could favor the formation of active oxygen species.

Petroleum refinery wastewater has a complex composition that affected the applied CWAO treatment, rendering it inefficient. Other factors that could have influenced this process were the low catalyst loading and mild operating conditions. Furthermore, a combined CWAO process with another efficient treatment would certainly allow high mineralization of PRW.

## 5. Conclusions

AgCuAlCe, AuCuAlCe and NiCuAlCe prepared by a two-step method involving a redox method and CuAlCe synthesized by a wet impregnation method were tested as catalysts in the catalytic wet air oxidation of phenol from an aqueous solution and petroleum refinery wastewater under mild conditions. The redox method could promote a close interaction between two NP metals on the support surface; therefore, it could have synthesized an efficient low-cost catalyst, a transition–noble mixture bimetallic nanostructure. The evidence of alloy formation in BNPs of AgCuAlCe and AuCuAlCe was provided by ultraviolet–visible spectroscopy with diffuse reflectance, transmission electron microscopy and temperature-programmed reduction of H_2_. Furthermore, AgCuAlCe and AuCuAlCe showed improvements in their reducibility, in contrast to NiCuAlCe. The reducibility and dispersion degree had a close relationship, which could be measured via the particle size by TEM, SEM and H_2-_TPR profiles. The larger particles were found to be of silver, whereas smaller particles were found to be of gold via analysis of TEM images. The BNPs of AgCuAlCe and AuCuAlCe not only were alloy particles but also were well dispersed.

The addition of a noble metal (gold or silver) drastically improved the phenol mineralization process. TOC removal was higher for the bimetallic catalysts AgCuAlCe and AuCuAlCe compared to the monometallic copper. The higher activity and selectivity to CO_2_ of the bimetallic gold–copper and silver–copper-supported catalysts could be attributed to the alloy compound formation and the synergetic effect of the M–Cu interaction. In contrast, NiCuAlCe did not exhibit the same behavior in terms of phenol, PWR, COD and TOC conversion, probably because an alloy synergistic nanostructure could not be achieved.

The leaching of copper could be avoided through a real bimetallic NP system—in this case, a BNP alloy system with a synergistic real metal–metal interaction. There was a significant improvement in the reaction conditions at temperatures and pressures as mild as possible, due to the efficient results of the degradation of phenol (93% COD and 89% TOC) using AuCuAlCe at 120 °C and 10 bar of O_2_.

PRW has several recalcitrant and persistent compounds with a high concentration of COD, especially in crude oil; therefore, it has a complex composition. This is why the applied CWAO treatment was inefficient. Other variables that could have influenced the process were the low catalyst loading and mild operating conditions. These parameters should be optimized to obtain better degradation efficiency of pollutants in PR wastewater. Furthermore, a combined CWAO process with another efficient treatment would certainly allow high mineralization of PRW.

## Figures and Tables

**Figure 1 nanomaterials-11-02570-f001:**
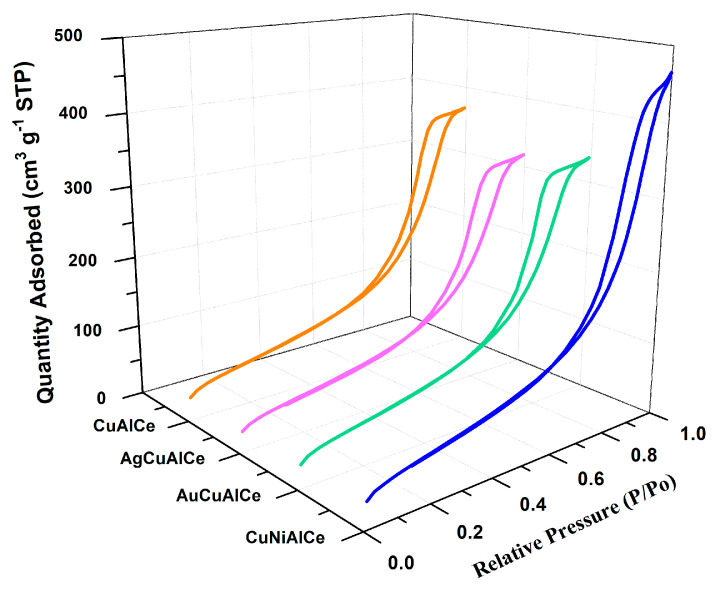
Adsorption–desorption isotherms for monometallic CuAlCe and bimetallic AgCuAlCe, AuCuAlCe and NiCuAlCe catalysts.

**Figure 2 nanomaterials-11-02570-f002:**
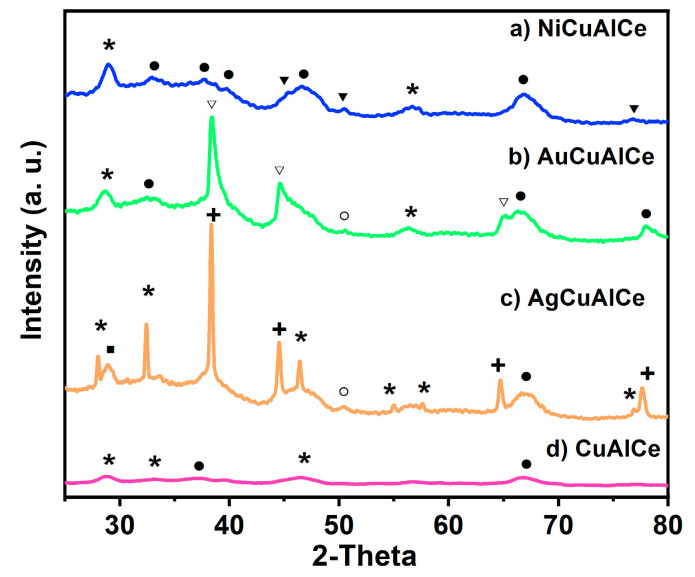
XRD patterns of (**a**) NiCuAlCe, (**b**) AuCuAlCe, (**c**) AgCuAlCe bimetallic-supported catalysts and (**d**) CuAlCe monometallic-supported catalyst, (+) Ag, (○) Cu, (∎) Cu_2_O, (▽) Au, (▼) Ni, (*) Ce, (⚫) Al.

**Figure 3 nanomaterials-11-02570-f003:**
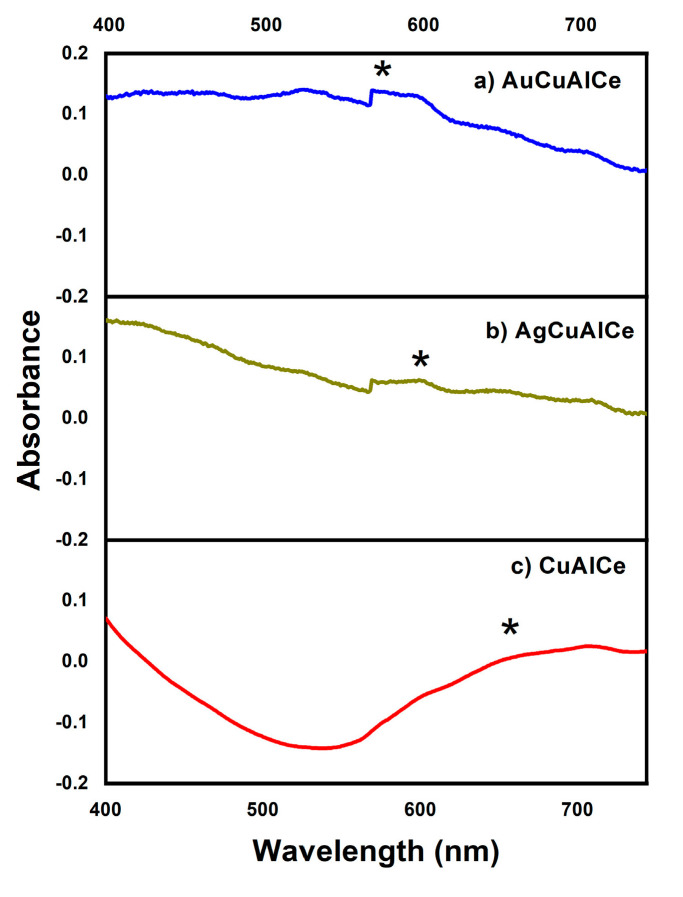
DR-UV-Vis spectra of AuCu and AgCu catalysts (**a**,**b**) and Cu-supported monometallic catalyst (**c**). Surface Plasmon Resonance: (*) SPR.

**Figure 4 nanomaterials-11-02570-f004:**
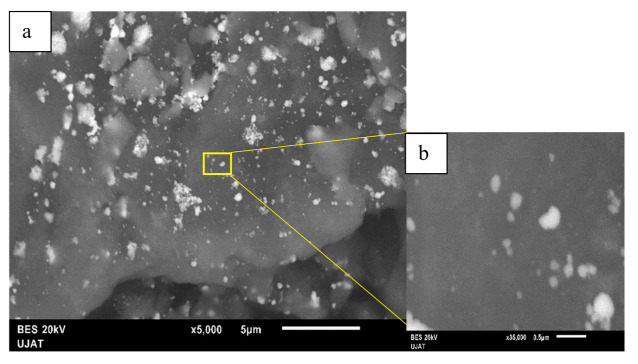
SEM images of AgCuAlCe (**a**) 5000× and (**b**) 35,000×.

**Figure 5 nanomaterials-11-02570-f005:**
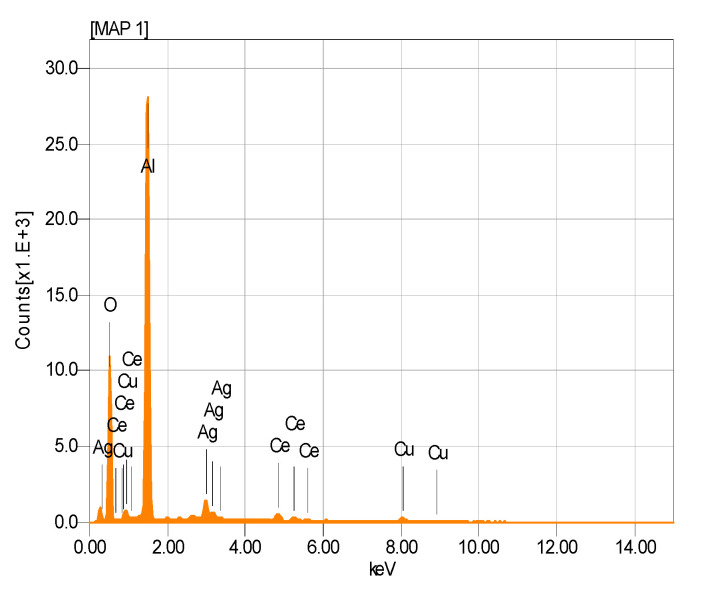
EDS spectrogram of AgCuAlCe bimetallic-supported catalyst.

**Figure 6 nanomaterials-11-02570-f006:**
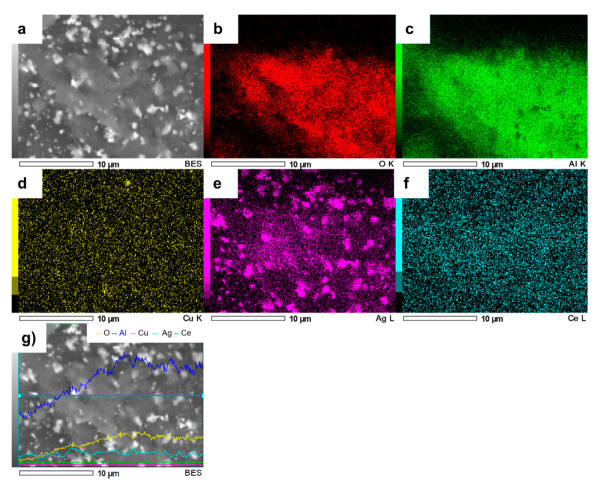
SEM image of AgCuAlCe (**a**) and the elemental mapping images of (**b**) O, (**c**) Al, (**d**) Cu, (**e**) Ag, (**f**) Ce and (**g**) EDS line scan of AgCuAlCe.

**Figure 7 nanomaterials-11-02570-f007:**
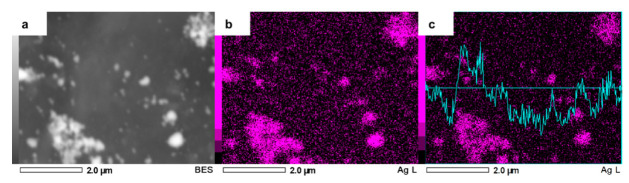
Zoomed-in SEM image of AgCuAlCe bimetallic catalyst (**a**) and the elemental mapping images of (**b**) Ag and (**c**) EDS line scan of Ag.

**Figure 8 nanomaterials-11-02570-f008:**
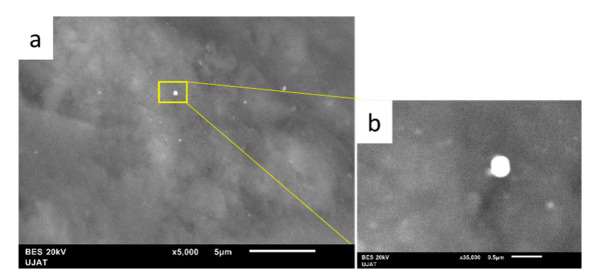
SEM images of NiCuAlCe bimetallic catalyst (**a**) 5000× and (**b**) 35,000×.

**Figure 9 nanomaterials-11-02570-f009:**
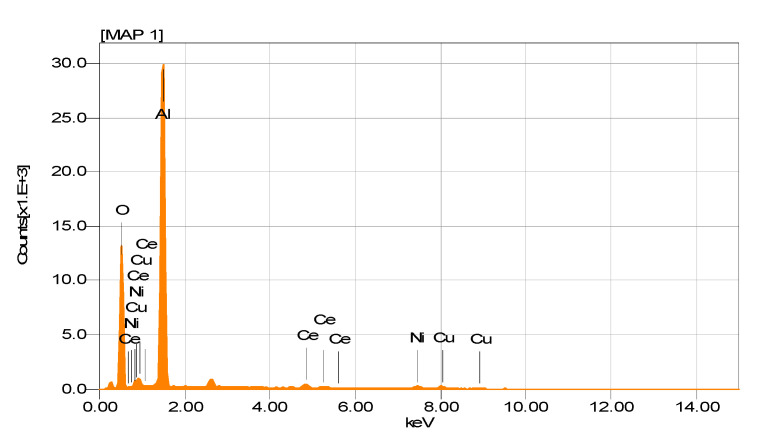
EDS spectrogram of NiCuAlCe bimetallic-supported catalyst.

**Figure 10 nanomaterials-11-02570-f010:**
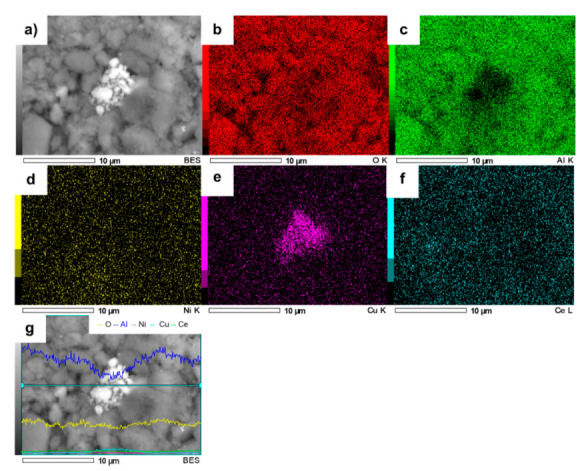
SEM image of NiCuAlCe (**a**) and the elemental mapping images of (**b**) O, (**c**) Al, (**d**) Ni, (**e**) Cu, (**f**) Ce and (**g**) EDS line scan of NiCuAlCe.

**Figure 11 nanomaterials-11-02570-f011:**
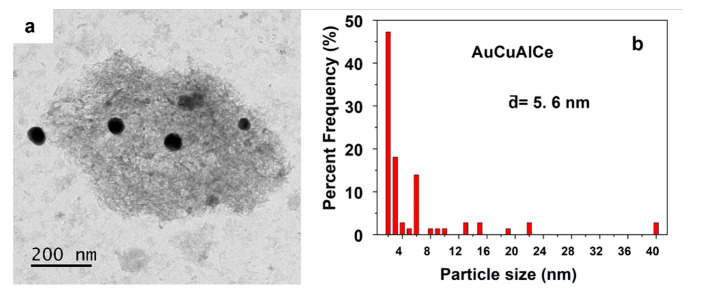
Particle size observed by (**a**) TEM imagen and (**b**) statistical distribution, of fresh AuCuAlCe bimetallic catalyst.

**Figure 12 nanomaterials-11-02570-f012:**
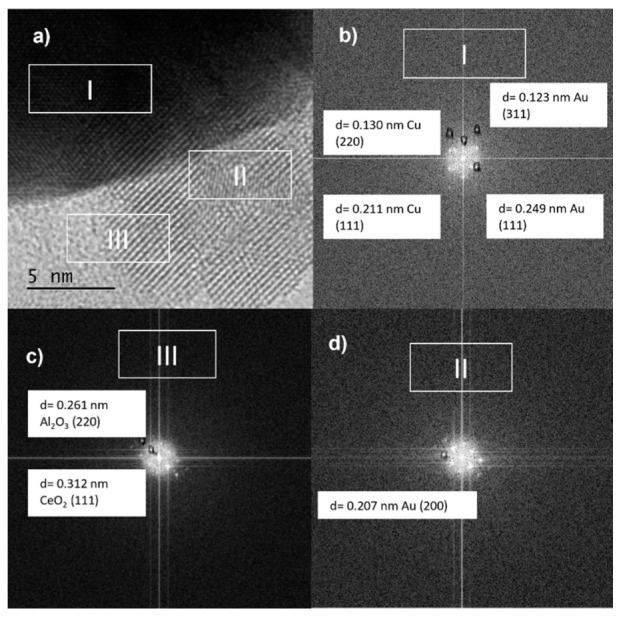
(**a**) TEM image and (**b**–**d**) Lattice spacing of arbitrarily selected bimetallic nanoparticles of bimetallic AuCuAlCe catalyst.

**Figure 13 nanomaterials-11-02570-f013:**
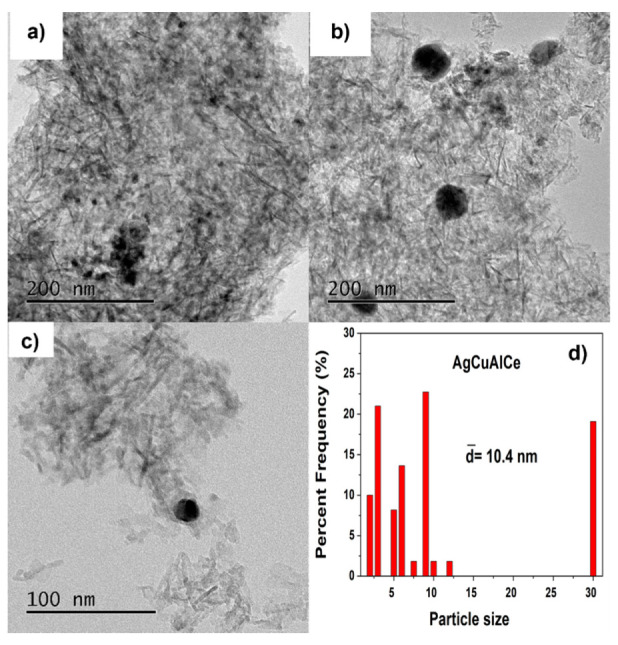
Particle size observed by TEM images of various surface regions at 200 (**a**,**b**) and 100 (**c**) nm. Statistical distribution (**d**), of fresh bimetallic AgCuAlCe catalyst.

**Figure 14 nanomaterials-11-02570-f014:**
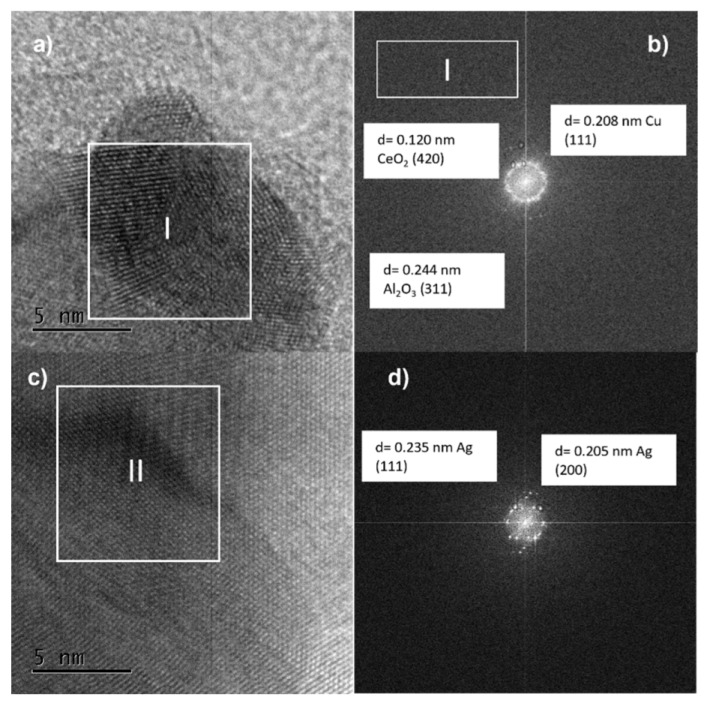
TEM images at 5 nm (**a**,**b**) and lattice spacing of arbitrarily selected nanoparticles (**c**,**d**) of fresh AgCuAlCe bimetallic catalyst.

**Figure 15 nanomaterials-11-02570-f015:**
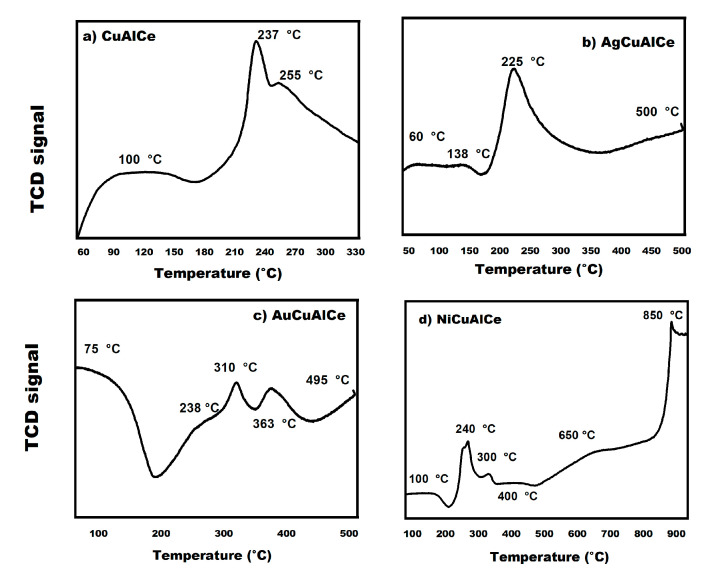
H_2_-TPR of (**a**) CuAlCe monometallic catalyst and (**b**) AgCuAlCe, (**c**) AuCuAlCe and (**d**) NiCuAlCe bimetallic-supported catalysts.

**Figure 16 nanomaterials-11-02570-f016:**
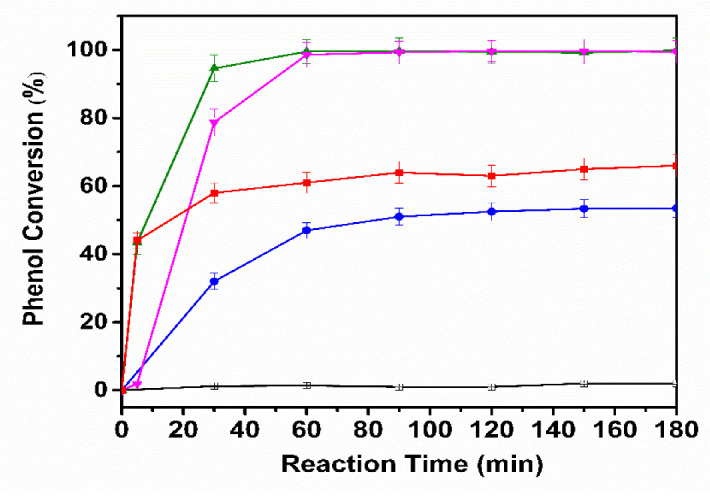
Evolution of the phenol concentration as a function of time upon the CWAO of phenol over monometallic CuAlCe (
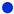
), bimetallic catalysts NiCuAlCe (
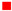
), AuCuAlCe (
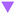
), AgCuAlCe (
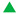
) and without catalyst (□).

**Figure 17 nanomaterials-11-02570-f017:**
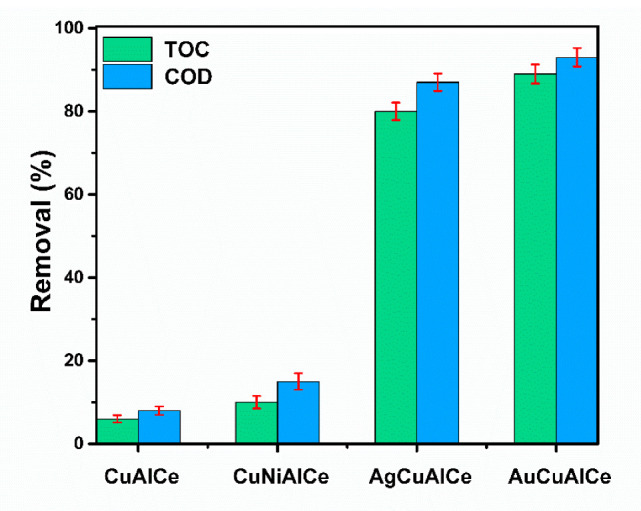
COD and TOC removal efficiency of monometallic CuAlCe and bimetallic catalysts NiCuAlCe, AuCuAlCe, AgCuAlCe obtained after 180 min of reaction. Operation conditions T = 120 °C, P(O_2_) = 10 bar, V_Liq_ = 0.25 L, Cfenol = 1000 mg L^−1^, C_Cat_ = 1 g L^−1^, ω = 1000 rpm.

**Figure 18 nanomaterials-11-02570-f018:**
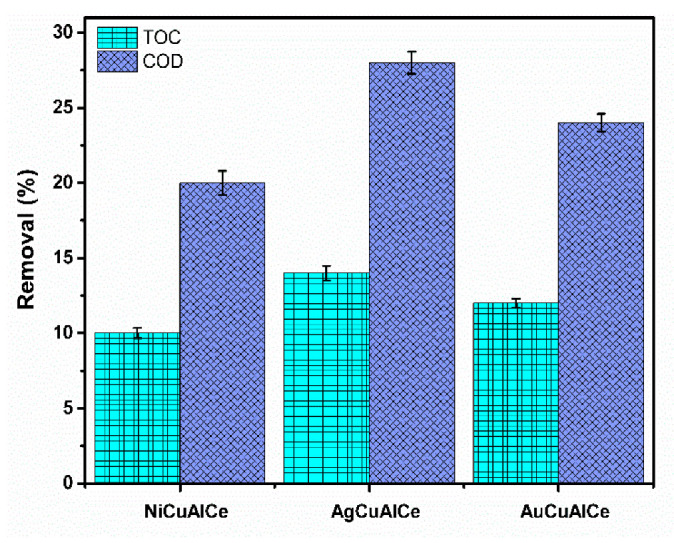
COD and TOC removal efficiency of petroleum refinery over bimetallic catalysts NiCuAlCe, AuCuAlCe, AgCuAlCe obtained after 180 min of reaction. Operation conditions T = 120 °C, P(O_2_) = 10 bar, V_Liq_ = 0.25 L, C_Cat_ = 1 g L^−1^, ω = 1000 rpm.

**Figure 19 nanomaterials-11-02570-f019:**
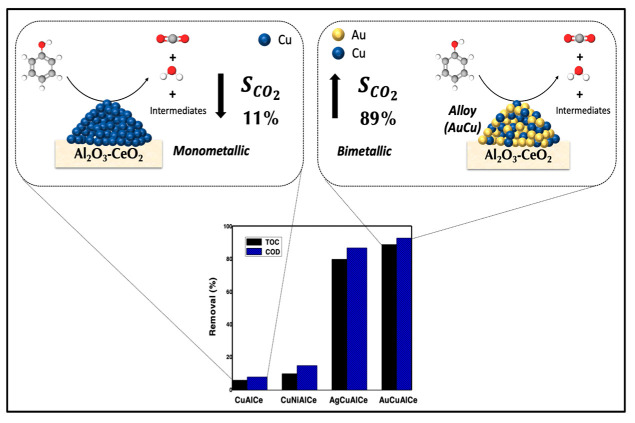
Scheme proposed to compare catalytic performance of phenol degradation with monometallic CuAlCe versus bimetallic catalyst AuCuAlCe.

**Table 1 nanomaterials-11-02570-t001:** Specific surface area (*S_BET_*), pore volume (P_V_), pore size (P_S_), crystallite size of oxide ($d_{S_{BET}}$, d_XRD_) of the prepared supports and catalysts.

Catalyst	*S_BET_* ^a^ (m^2^g^−1^)	P_V_ ^a^ (cm^3^g^−1^)	P_S_ ^a^ (nm)	$d_{S_{BET}}$^a^ (nm)	d_XRD_, CeO_2_ ^b^ (nm)	d_XRD_, Al_2_O_3_ ^b^ (nm)
AlCe	321	1.880	16	1.7	-	-
CuAlCe	268	0.839	12	2	3	4
AgCuAlCe	191	0.490	8	3	11	-
AuCuAlCe	211	0.520	7	2.6	5	7
NiCuAlCe	236	0.697	9	2.4	4	2

^a^ Determined by N_2_ adsorption–desorption isotherms. ^b^ Calculated from XRD data, with Scherrer’s equation, (222) crystallographic plane of CeO_2_ and (440) crystallographic plane of Al_2_O_3_.

**Table 2 nanomaterials-11-02570-t002:** Metallic particle size of bimetallic catalysts.

Bimetallic Catalyst	Metallic Particle by Scherrer’s Equation (nm) ^a^	Metallic Particle by SEM (nm)
AgCuAlCe	14	36
AuCuAlCe	9	n. d.
NiCuAlCe	n. d.	74

^a^ Calculated from XRD data, with Scherrer´s equation, (111) crystallographic plane of Ag and (111) crystallographic plane of Au. n. d. not detected.

**Table 3 nanomaterials-11-02570-t003:** Activity for the catalytic wet air oxidation of phenol after 180 min of reaction. Phenol conversion (X_C_), COD removal (X_COD_), TOC removal (X_TOC_), initial rate (r_i_), intermediate concentration of catechol (A) and CO_2_ Selectivity (S_CO2_) as a function of time for bimetallic copper-based and monometallic copper catalysts.

Catalysts	X_C_ ^a^ (%)	X_COD_ ^a^ (%)	X_TOC_ ^a^ (%)	A (mmol/L)	r_1_ ^a^ (mmol h^−1^g^−1^ _phenol_)	S_CO2_
CuAlCe	54	8	6	27	681	11
AgCuAlCe	100	87	80	10	2022	80
AuCuAlCe	99	93	89	9	1680	89
NiCuAlCe	66	15	10	22	1234	15

^a^ Obtained after 180 min of reaction.

**Table 4 nanomaterials-11-02570-t004:** Properties of industrial wastewater sample.

pH	Temperature (°C)	COD (mg·L^−1^)	TOC (mg·L^−1^)
3.5	38–40	1909	4720

## Data Availability

Not applicable.

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
