# Peer review of "Bimetallic M–Cu (M = Ag, Au, Ni) Nanoparticles Supported on γAl2O3-CeO2 Synthesized by a Redox Method Applied in Wet Oxidation of Phenol in Aqueous Solution and Petroleum Refinery Wastewater"

_nanomaterials, 2021, doi:10.3390/nano11102570_

Round 1

Reviewer 1 Report

This manuscript reports a preparation method of a bimetallic catalyst used to catalyze phenol in refinery wastewater. Three kinds of M-copper bimetallic catalysts supported by M=silver, gold and nickel were successfully prepared by a two-step synthesis method. Based on the synergistic effect of the two, the activity and stability of the catalyst were improved. It is worthy of recognition that the author has done a lot of literature research and experimental work. However, there is lack of adequate materials to meet the full article type. Thus, I think this manuscript could be accepted after a minor revision.

1. The manuscript introduces in the preface that the high price of precious metals restricts the use of renewable energy in CWAO, but a lot of work has been done on precious metals. In order to keep the logic complete, the reasons for this part of the work need to be introduced in this section. Compared with precious metals, what are the advantages of transition metal Ni? 2. What is the cause of Cu aggregation in Figure 10? Why is there no uniform distribution? Please explain the results of the experiment. 3. It is recommended to improve the quality of Figure 4 and Figure 8, and mark the available size in the figure. 4. The author needs to carefully proofread the format of the article and make corrections to the following three points. ".," on line 114? "120 °C and 10 bar O2" on line 122? "Cu0" in line 322? 5. The entire manuscript needs English polishing one more time.

Author Response

Dear reviewer 1:

Please, find the corrections suggested to the manuscript, entitled “Bimetallic  M-Cu (M=Ag, Au and Ni) Nanoparticles supported: Effect of the second metal on catalytic performance over Wet Oxidation of Phenol in aqueous solution and Petroleum Refinery Wastewater” by Zenaida Guerra Que, Jorge Cortez Elizalde, Hermicenda Pérez Vidal, , Juan Carlos Arévalo Pérez, Adib Abiu Silahua-Pavón, Gerardo E. Córdova Pérez, Ignacio Cuauhtémoc López, Héctor Martínez García, Anabel González Díaz and José Gilberto Torres Torres. Thank a lot for your recommendations. They were an excellent guideline to improve the manuscript.

Reviewer 2 Report

It is a very good study with overall adequate presentation of experimental results. Some additions are needed:

1) Authors should further emphasize on the novelty of their work.

2) Some minor typos, grammar and syntax errors should be carefully revised and corrected accordingly.

3) Reference can be even more updated (more recent relative works).

4) Figs 16-18: Standard deviation is need to add.

Author Response

Villahermosa, Tab., México; June 26, 2021

Dear reviewer 2:

Please, find the corrections suggested to the manuscript, entitled “Bimetallic  M-Cu (M=Ag, Au and Ni) Nanoparticles supported: Effect of the second metal on catalytic performance over Wet Oxidation of Phenol in aqueous solution and Petroleum Refinery Wastewater” by Zenaida Guerra Que, Jorge Cortez Elizalde, Hermicenda Pérez Vidal, , Juan Carlos Arévalo Pérez, Adib Abiu Silahua-Pavón, Gerardo E. Córdova Pérez, Ignacio Cuauhtémoc López, Héctor Martínez García, Anabel González Díaz and José Gilberto Torres Torres. Thank a lot for your recommendations. They were an excellent guideline to improve the manuscript.

Round 2

Reviewer 2 Report

All my comments of the initial submission have been correctly replied and included in the revised manuscript. The quality of this work has been drastically improved after revision and therefore I recommend its publication as it is.